# `VQ-TR`: Vector Quantized Attention for Time Series Forecasting

**Kashif Rasul, Andrew Bennett, Pablo Vicente, Anderson Schneider & Yuriy Nevmyvaka**
Morgan Stanley, New York, USA
`kashif.rasul@gmail.com`

**Umang Gupta**
USC, Los Angeles, USA

**Hena Ghonia**
Université de Montréal, Montréal, Canada

## Abstract

Probabilistic time series forecasting is a challenging problem due to the long sequences involved, the large number of samples needed for accurate probabilistic inference, and the need for real-time inference in many applications. These challenges necessitate methods that are not only accurate but computationally efficient. Unfortunately, most current state-of-the-art methods for time series forecasting are based on Transformers, which scale poorly due to quadratic complexity in sequence length, and are therefore needlessly computationally inefficient. Moreover, with a few exceptions, these methods have only been evaluated for non-probabilistic point estimation. In this work, we address these two shortcomings. For the first, we introduce `VQ-TR`, which maps large sequences to a discrete set of latent representations as part of the Attention module. This not only allows us to attend over larger context windows with linear complexity in sequence length but also allows for effective regularization to avoid overfitting. For the second, we provide what is to the best of our knowledge the first systematic comparison of modern Transformer-based time series forecasting methods for probabilistic forecasting. In this comparison, we find that `VQ-TR` performs better or comparably to all other methods while being computationally efficient.

## 1 Introduction

Time series forecasting is a challenging machine learning task given the need to model complex, non-linear temporal patterns over potentially long time horizons. Recently, methods based on Transformers (Vaswani et al., 2017) have dominated the state-of-the-art, outperforming both classical autoregressive approaches, as well as deep learning approaches using Convolutional Neural Networks (CNNs) or Recurrent Neural Networks (RNNs). This is in large part due to the Transformer's strong inductive bias (Zhou et al., 2021), which allows it to look back over the entire context history of a time series, without suffering from limited temporal field or issues of forgetting (Mahto et al., 2021).

Unfortunately, the recent line of work on using transformers for time series forecasting has two major limitations. First, the proposed methods generally use very computationally inefficient architectures, given the quadratic complexity in sequence length necessitated by the self-attention mechanism. This problem is most pronounced in problems where sequence lengths are long, although even with medium sequence lengths it results in unnecessarily inefficient computation. In particular, these problems are exacerbated in probabilistic time series forecasting settings, where many sample trajectories must be generated, and forecasting must often be performed in real time.

A second issue with the past work on transformer-based time series forecasting is that, with very few exceptions,[1] these methods have only been evaluated in terms of their performance for *point forecasting* (that is, predicting the mean values of future sequences), rather than their performance for *probabilistic forecasting* (that is, predicting the full probability distribution of future sequences). This is unfortunate, since for many applications areas probabilistic forecasting is paramount. Although arguably it is straightforward to use these methods to perform probabilistic forecasting by replacing the heads of the models with probabilistic emission heads, there are no existing standard benchmarks for comparing the performance of different transformer architectures for this task.

In this paper, we address these shortcomings via a two-pronged approach. For the first, we propose a novel Transformer architecture for time series forecasting, `VQ-TR`, which eliminates the quadratic complexity in sequence length of attention via a Vector Quantization (van den Oord et al., 2017) module. For the second, we provide a systematic comparison of the performance of an extensive set of Transformer-based methods for probabilistic time series forecasting on a wide range of datasets, in terms of many different performance metrics for probabilistic inference. In this comparison, we find that `VQ-TR` performs better or comparably to all other methods while being significantly more computationally efficient. In addition, we find that the Vector Quantization module not only improves computational and memory efficiency but also improves forecasting performance due to a natural regularizing effect.

## 2 BACKGROUND

### 2.1 PROBABILISTIC TIME SERIES FORECASTING

The task of probabilistic time series forecasting in the *univariate* setting consists of training on a dataset of $D \geq 1$ time series $\mathcal{D}_{\text{train}} = \{x^i_{1:T^i}\}$ where $i \in \{1, \ldots, D\}$ and at each time point $t$, we have $x^i_t \in \mathbb{R}$ or $\mathbb{N}$. We are tasked with predicting the potentially complex distribution of the next $P > 1$ time steps into the future, and we are given a test set $\mathcal{D}_{\text{test}} = \{x^i_{T^i+1:T^i+P}\}$. Each time index $t$ is in practice a date-time value that increments regularly based on the frequency of the dataset in question, and the last training point $T^i$ for each time series may or may not be the same date-time. Autoregressive models like those in Graves (2013) or Salinas et al. (2019b) estimate the prediction density by decomposing the joint distribution of all $P$ points via the chain rule of probability as:

$$p_{\mathcal{X}}(x^i_{T^i+1:T^i+P}) \approx \Pi^P_{t=1} p(x^i_{T^i+t} | x^i_{1:T^i-1+t}, \mathbf{c}^i_{1:T^i+P}; \theta),$$

parameterized by some model with trained weights $\theta$. This requires the next time point being conditioned on *all* the past and covariates $\mathbf{c}^i_t$ (detailed in Section 3.3), which is computationally challenging to scale, especially if the time series has a considerably long history. Models like `DeepAR` (Salinas et al., 2019b) typically resort to the seq-to-seq paradigm (Sutskever et al., 2014) and consider some context window of *fixed-size $C$* sampled randomly from the complete time series history to learn some historical representation and use this representation in the decoder to learn the distribution of the subsequent time points of the context. This does, however, mean that the model falls short of capturing seasonalities in its prediction, which can lead to a worse approximation of the future distribution.

Encoder-decoder Transformers (Vaswani et al., 2017) naturally fit the seq-to-seq paradigm, where $N$ encoding Transformer layers can be used to learn a size $C$ sequence of representations, denoted by:

$$\{\mathbf{h}_t\}^{C-1}_{t=1} = \text{Enc}_N \circ \cdots \circ \text{Enc}_1(\{\texttt{concat}(x^i_t, \mathbf{c}^i_{t+1})\}^{C-1}_{t=1}; \theta).$$

Afterward, $M$ layers of a *causal* or masked decoding Transformer can be used to model the subsequent $P$ future time points conditioned on the encoding representations as:

$$\Pi^{C+P-1}_{t=C} p(x^i_{t+1} | x^i_{t:C}, \mathbf{c}^i_{t+1:C+1}, \mathbf{h}_1, \ldots, \mathbf{h}_{C-1}; \theta).$$

For example, if we assume the data comes from a Student-T distribution then the outputs of the Transformer's $M$ decoders can be passed to a layer that returns appropriately signed parameters of a

---

[1]Of note, Lim et al. (2021a) does probabilistic forecasting with transformer architectures via quantile regression.

Student-T distribution. Then, we can maximize the log-likelihood of the resulting Student-T given the predicted parameters given by

$$\sum_{t=C}^{C+P-1} \log p_{\mathcal{T}}(x_{t+1}^i | x_{t:C}^i, \mathbf{c}_{t+1:C+1}^i, \mathbf{h}_1, \ldots, \mathbf{h}_{C-1}; \theta),$$

for all $i, t$ from $\mathcal{D}_{\text{train}}$ using stochastic gradient descent (SGD), as described in Section 3.1. Note that this approach can be used with *any* choice of distribution class, not just student-T, with the Transformer's decoders returning the parameter values for the chosen distribution class.

Transformers offer a viable alternative to recurrent neural networks (RNN), such as LSTM (Hochreiter and Schmidhuber, 1997) or GRU (Chung et al., 2014), which, apart from being sequential, suffer from forgetting for large context windows, or Convolutional models like TCN (Bai et al., 2018), which have limited temporal receptive fields. However, transformers scale quadratically with sequence length in the compute and memory *per* layer. Reducing the computational requirements of Transformers is an active area of research, and several strategies have been proposed, for example, by compressing the sequence (Wang et al., 2020), exploiting locality (Beltagy et al., 2020), or mitigating computation for each of the input entities (Hawthorne et al., 2022). In contrast, our approach works by quantizing the representations *without modifying the inputs*, using Vector Quantization as described next.

## 2.2 VECTOR QUANTIZATION (VQ)

The VQ-VAE (van den Oord et al., 2017; Razavi et al., 2019) is an encoder-decoder Variational Autoencoder (VAE) (Kingma and Welling, 2019) that encodes inputs onto a set of $J \geq 1$ discrete latent variables called the codebook $\{\mathbf{z}_1, \ldots, \mathbf{z}_J\}$. The decoder reconstructs the inputs from the resulting discrete vectors. The input vector is quantized with respect to its distance to its nearest codebook vector via the Vector Quantization (VQ) operator, which is defined by:

$$\text{Quantize}(\mathbf{q}) := \mathbf{z}_n, n \quad \text{where} \quad n = \arg\min_j \|\mathbf{q} - \mathbf{z}_j\|_2. \tag{1}$$

Due to the non-differentiable VQ operation, the codebook is learned by straight-through gradient estimation (Hinton et al., 2012; Bengio et al., 2013).[2]

Additionally, when using VQ one introduces two extra loss terms. First, we include a *latent loss* term, which encourages alignment of codebook vectors to the inputs of VQ, and therefore ensures that the straight-through gradient estimation is reasonable. Second, we include a *commitment loss* term, which encourages inputs to "commit" to particular code vectors, and heuristically helps with convergence behavior and preventing cycling. These are implemented via the "stop-gradient" (sg) or "detach" operators of deep learning frameworks, which block gradients from flowing into its argument. The additional VQ loss terms can be concretely written as:

$$\textbf{latent loss: } \|\text{sg}(\mathbf{q}) - \mathbf{z}\|_2^2 \qquad \textbf{commitment loss: } \beta\|\text{sg}(\mathbf{z}) - \mathbf{q}\|_2^2, \tag{2}$$

where $\beta$ is a hyperparameter weighting the commitment loss.

## 3 VQ-TR MODEL

We motivate our methodology for incorporating VQ within the transformer architecture with an observation about the effect of approximations of the query vector on self-attention. Recall that in self-attention, the incoming vector sequences are mapped to query, key, and value vectors, denoted by $\mathbf{q}_t, \mathbf{k}_t$, and $\mathbf{v}_t$, respectively, for each time step $t$. Let us denote the approximation of the query vector $\mathbf{q}_t$ by $\hat{\mathbf{q}}_t$. The attention weight for sequence index $t$ attending on some other index $u$ is

$$w_{tu} = \frac{\exp(\mathbf{q}_t^T \mathbf{k}_u)}{\sum_j \exp(\mathbf{q}_t^T \mathbf{k}_j)},$$

and the output representation are correspondingly given by, $\mathbf{o}_t = \sum_u w_{tu} \mathbf{v}_u$ (Phuong and Hutter, 2022). We then have the following (see Appendix A for proof):

---

[2]That is, the gradients coming from upstream of the VQ module are copied downstream. This is equivalent to incorrectly using the gradient at $\text{Quantize}(q)$ instead of the gradient at $q$ when performing backpropagation, and is a reasonable approximation when $\text{Quantize}(q) \approx q$.

**Theorem 3.1.** *If $\sum_u \left| \mathbf{q}_t^T \mathbf{k}_u - \hat{\mathbf{q}}_t^T \mathbf{k}_u \right| \leq \delta_t$ for sufficiently small $\delta_t > 0$, then the attention weights with respect to the approximation $\hat{\mathbf{q}}_t$, which we denote $\hat{w}_{tu}$, satisfy the bounds*

$$w_{tu}(1 - 2\delta_t) \leq \hat{w}_{tu} \leq w_{tu}(1 + 2\delta_t) \qquad \forall u.$$

*Further, as a result $|\mathbf{o}_t - \hat{\mathbf{o}}_t| \leq 2\delta_t \mathbf{o}_t$ holds element-wise for the output representation.*

The above result bounds the error in the output representation from the self-attention module due to errors in the query vector. It is clear from this result that we can ensure that the outputs have a good approximation on average for all queries if we pick codebook vectors $\{\mathbf{z}_1, \ldots, \mathbf{z}_J\}$ such that $\mathbb{E}_{\mathbf{q} \in \mathcal{Q}, \mathbf{k} \in \mathcal{K}}[\min_{j=1}^J \left| \mathbf{q}^T \mathbf{k} - \mathbf{z}_j^T \mathbf{k} \right|]$ is small, where the expectation operator $\mathbb{E}_{\mathbf{q} \in \mathcal{Q}, \mathbf{k} \in \mathcal{K}}$ corresponds to averaging over all queries and keys. Now, by Jensen's inequality, the squared of this average is bounded by $\mathbb{E}_{\mathbf{q} \in \mathcal{Q}, \mathbf{k} \in \mathcal{K}}[\min_{j=1}^J (\mathbf{q}^T \mathbf{k} - \mathbf{z}_j^T \mathbf{k})]^2 = \mathbb{E}_{\mathbf{q} \in \mathcal{Q}, \mathbf{k} \in \mathcal{K}} \min_{j=1}^J (\mathbf{q}^T \mathbf{k} - \mathbf{z}_j^T \mathbf{k})^2$. We can minimize a looser upper bound for these output representation errors by minimizing this mean squared error. This has a nice interpretation, since

$$\mathbb{E}_{\mathbf{q} \in \mathcal{Q}, \mathbf{k} \in \mathcal{K}}\left[ \min_{j=1}^J (\mathbf{q}^T \mathbf{k} - \mathbf{z}_j^T \mathbf{k})^2 \right] = \mathbb{E}_{\mathbf{q} \in \mathcal{Q}, \mathbf{k} \in \mathcal{K}}\left[ \min_{j=1}^J (\mathbf{q} - \mathbf{z}_j)^T \mathbf{k} \mathbf{k}^T (\mathbf{q} - \mathbf{z}_j) \right]$$

$$= \mathbb{E}_{\mathbf{q} \in \mathcal{Q}}\left[ \min_{j=1}^J (\mathbf{q} - \mathbf{z}_j)^T \left( \boldsymbol{\Sigma}_k + \boldsymbol{\mu}_k \boldsymbol{\mu}_k^T \right)(\mathbf{q} - \mathbf{z}_j) \right],$$

which is a $K$-means objective weighted by $\boldsymbol{\Sigma}_k + \boldsymbol{\mu}_k \boldsymbol{\mu}_k^T$, where $\boldsymbol{\mu}_k$ and $\boldsymbol{\Sigma}_k$ denote the mean and covariance of the key vectors respectively. Thus, given that the optimal codes of VQ are known to be the $K$-means clusters of the input representations (van den Oord et al., 2017), this suggests VQ is a natural approach to ensure small output representation errors.

Given this motivation, in the `VQ-TR` model, we modify the Transformer's encoder architecture by first mapping the $C$ incoming query vectors, denoted by $Q \in \mathbb{R}^{C \times F}$, through a VQ module:

$$Z_0, \text{indices} := \text{VQ}(Q).$$

where $Z_0 \in \mathbb{R}^{J \times F}$ denotes the latent codebook vectors, and indices $\in \{1, \ldots, J\}^C$ denotes the sequences of $C$ indices from these $J$ codebook vectors. Next, we apply cross-attention of the keys/values with the latent codebook vectors to obtain $Z_1 \in \mathbb{R}^{J \times F}$:

$$Z_1 := \text{CrossAttn}(Z_0, K, V),$$

and then process them further via self-attention $L$ times: $Z_{l+1} := \text{SelfAttn}(Z_l)$. Finally, we return the original number of sequences by gathering the resulting latent vectors via the indices with respect to the quantization of the input vectors $Q$:

$$\mathbb{R}^{C \times F} \ni Z := \text{Gather}(Z_{L+1}, \text{indices}).$$

This total construction can be repeated $N$ times, allowing for $N$ encoding layers. Since there are only $J$ latent vectors, and in practice $J \ll C$, we obtain an architecture with memory and compute complexity of $O(CJ)$ and $O(LJ^2)$ from the cross-attention and latent self-attention respectively (Jaegle et al., 2021; Hawthorne et al., 2022) per number of encoding layers $N$. This is in contrast to memory and compute complexity of $O(LC^2)$ for each attention layer with the vanilla Transformer.

One downside to this architecture is that we lose the ability to impose a causal structure. Fortunately, we can resolve this problem by first using the above non-causal encoder to learn discrete representations of large context windows, and then using a *causal* Transformer decoder to predict future time series values, which is a standard kind of approach for time series forecasting. Unfortunately, given $M$ decoding layers, the decoder will then scale as $O(MP^2)$. On the other hand, in practice (including in our experiments) we often have $P \ll C$, which mitigates this downside, so we can still train and perform inference efficiently conditioned on long histories. We present a schematic of the `VQ-TR` model in Figure 1 for both the training (Section 3.1) and inference (Section 3.2) scenarios.

### 3.1 TRAINING

Given a set $\mathcal{D}_{\text{train}}$ of $D \geq 1$ time series, we construct batches of inputs by randomly sampling univariate time series $\{x_{1:T^i}^i\}$, with $i \in \mathbb{Z}^+$ such that $i \leq D$, then selecting random $t \in \mathbb{Z}^+$ with

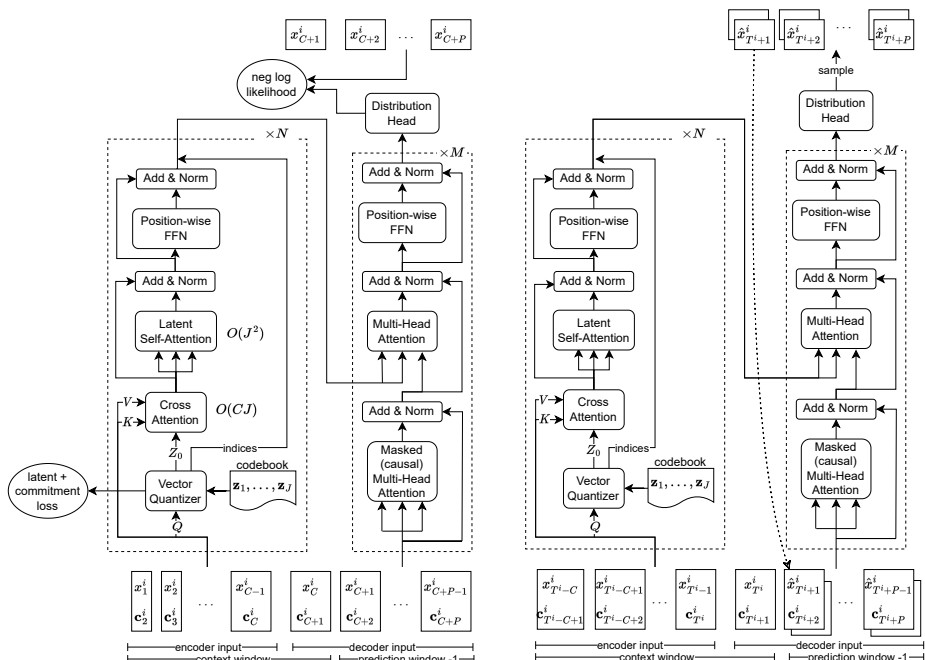

Figure 1: VQ-TR model with $N$ encoding vector-quantized cross-attention blocks and $M$ *causal* decoding transformer blocks. During training (left), the encoder takes a sequence of length $C - 1$ and the decoder outputs the estimated distribution parameters (of some chosen distribution class) for the next $P$ time steps, which are learned via the negative log-likelihood together with the $N$ Vector Quantizer losses. During inference (right), we pass the last $C - 1$ length context window to the encoder and the very last value in it to the decoder, which allows us to sample the next time step which we autoregressively pass back to the decoder to obtain predictions for the desired horizon.

$t \leq T^i - C - P$, which gives us context window $\{x_{t:t+C}^i\}$ and corresponding prediction window $\{x_{t+C+1:t+C+P}^i\}$, for fixed context and prediction window lengths $C$ and $P$ respectively.

For each batch step, we jointly minimize the negative log-likelihood of the predicted distribution with respect to the ground truth predictions, along with the $N$ latent and commitment losses from the VQ module placed on each encoder layer. This is in contrast to the practice of first learning the discrete latent representations in an unsupervised fashion and then using these latent representations for downstream tasks, as in, for example, the DALL·E model (Ramesh et al., 2021).

Note, VQ-TR models each time series independently, using a shared model. For example, if we have a multivariate dataset with $K$ time series, the future values for each of those time series are assumed to depend only on past values of that specific time series, and not on any values of the other $K - 1$ time series. However, we do use a common model for the relationship between past and future values, which is applied independently to each of the $K$ time series. Also, note that this approach is used for all models in our comparison, not just VQ-TR.

## 3.2 INFERENCE

During inference, we feed the *final* $C$-sized window (minus the last entry) for each time series $i \in \mathcal{D}_{\text{train}}$ to the encoder, and the last entry to the decoder, to obtain estimated distribution parameters of the next entry. We then sample one or more values from this distribution, and feed these back to the decoder. Repeating this $P$ times gives us sample trajectories over the prediction window.

Note that we only need to run the encoder once to make predictions, and we can repeat tensors in the batch dimension to obtain multiple samples from the distribution in parallel. If a point forecast is required, we can evaluate the empirical mean or median at each time point of the prediction.

## 3.3 COVARIATES

Positional encoding allows the Transformer to encode positional information of sequences, which is necessary since Attention is a permutation equivariant layer. In the time series setting, we can naturally create positional encodings like Rotary Positional Embedding (RoPE) (Su et al., 2021) via date-time covariates. More specifically, for a particular time point $t$, depending on the frequency of the time series $i$, we can create hour-of-day, day-of-week, week-of-month, etc. features as a vector, which we denote by $\mathbf{c}_t^i$. Due to their deterministic temporal nature, we can build these covariates for all future time points we wish to forecast. Additional covariates can be constructed by embedding the identity $i$ of each time series in a dataset via Embedding layers, as done in the `DeepAR` method. *All* the methods considered receive the same covariates as input.

## 3.4 SCALING

Time series data can be of an arbitrary numerical magnitude within a dataset. This is unlike vision, NLP, or even audio modalities. So to train a shared model over potentially very different time series, we calculate the mean value of the signal within its context window and divide the signal with it to normalize (Salinas et al., 2019b). The scale value of the context is kept as a covariate. More importantly, the model's output distribution is transformed back to the original scale, using the scale value from training or inference, to respectively calculate log probabilities or sample next points. If the scaling cannot be done in the output distribution's parameter space, one can do it in the data space after sampling. *All* deep learning-based methods in Section 4 incorporate this heuristic.

# 4 EXPERIMENTS

## 4.1 DATASETS

We use the following open datasets: `Exchange` (Lai et al., 2018), `Solar` (Lai et al., 2018), `Elecricity`[3], `Traffic`[4], `Taxi`[5], and `Wikipedia`[6] preprocessed exactly as in Salinas et al. (2019a). The properties of the dataset are summarized in Table 3 of the Appendix. These datasets cover a range of time series domains, including finance, weather, energy, logistics, and page views. We also note that we do not normalize scales for the traffic dataset, as its values are bounded.

## 4.2 EVALUATION CRITERIA

We evaluate all methods both in terms of their performance for point forecasting and also for probabilistic forecasting. As emphasized in Section 1, this is to the best of our knowledge the first extensive, systematic comparison of transformer-based methods for probabilistic forecasting.

**Point forecasting metrics:** We use the normalized root mean square error (NRMSE), the mean absolute scaled error (MASE) (Hyndman and Koehler, 2006), and the symmetric mean absolute percentage error (sMAPE) (Makridakis, 1993). For these metrics, we use the sample *median* except for NRMSE, where we report sample *mean*.

**Probabilistic forecasting metrics:** We follow the recommendations of the M4 competition (Makridakis et al., 2020) for evaluating probabilistic forecasting. Specifically, we report the mean scale interval score (MSIS[7]) (Gneiting and Raftery, 2007) for the 95% prediction interval, the 50th and 90th quantile percentile loss (QL50 and QL90, respectively), as well as the continuous ranked probability score (CRPS) (Gneiting and Raftery, 2007; Matheson and Winkler, 1976). The CRPS is a *proper* scoring rule that measures the compatibility of a predicted cumulative distribution function (CDF) $F$

---

[3]https://archive.ics.uci.edu/ml/datasets/ElectricityLoadDiagrams20112014
[4]https://github.com/laiguokun/multivariate-time-series-data#traffic-usage which is *not* the same dataset as in `TFT` Lim et al. (2021b).
[5]https://www1.nyc.gov/site/tlc/about/tlc-trip-record-data.page
[6]https://github.com/mbohlkeschneider/gluon-ts/tree/mv_release/datasets
[7]http://www.unic.ac.cy/test/wp-content/uploads/sites/2/2018/09/M4-Competitors-Guide.pdf

with the ground-truth samples $x$ as

$$\mathrm{CRPS}(F, x) = \int_{\mathbb{R}} (F(y) - \mathbb{I}\{x \le y\})^2 \, \mathrm{d}y,$$

where $\mathbb{I}\{x \le y\}$ is 1 if $x \le y$ and 0 otherwise. We approximate the CDF via empirical samples at each time point, and the final metric is averaged over all context windows of the time series, and time steps within the corresponding prediction window.

### 4.3 BASELINES

We compare `VQ-TR` against the Vanilla `Transformer`, as well as the following additional baselines, whose details can be found in the Appendix:

**Transformer-based Baselines:** `TFT` (Lim et al., 2021b), `Informer` (Zhou et al., 2021), `Autoformer` (Wu et al., 2021), `ETSformer` (Woo et al., 2022), `Hopfield` (Ramsauer et al., 2021), `Longformer` (Beltagy et al., 2020), `Reformer` (Kitaev et al., 2020), `Linformer` (Wang et al., 2020), `Nystromformer` (Xiong et al., 2021), `Performer` (Choromanski et al., 2021) and `PatchTST` (Nie et al., 2023).

**Other Deep-learning Baselines:** `DeepAR` (Salinas et al., 2019b), `MQCNN` (Wen et al., 2017), `SQF-RNN` (Gasthaus et al., 2019), `IQN-RNN` (Gouttes et al., 2021), `VQ-AR` (Rasul et al., 2022), and `D-Linear` (Zeng et al., 2023).

**Classical Baseline:** `ETS` (Hyndman and Khandakar, 2008).

### 4.4 RESULTS

We detail the results of our extensive experiments for `VQ-TR` and the transformer-based baselines in Table 1. In addition, we detail results for the other less competitive baselines in the Appendix.[8] As can be seen in Table 1, `VQ-TR`, performs competitively compared to all other transformer-based methods. In particular, it performs best on almost all metrics on 5 out of 6 datasets. Moreover, the performance is close to the best-performing method in cases where it does not outperform. We observe similar behavior as well against the non-transformer baselines in Table 2 in Appendix B, where we also provide a more detailed discussion of the relative performance of VQ-TR. We hypothesize that this competitive performance is in part due to the vector quantization module, which may function as a noisy channel similar to a VAE, regularizing the learning naturally and improving test performance. Indeed, we provide some positive evidence for this in our vector quantization ablation.

Next, we investigate the computational efficiency of the `VQ-TR` transformer architecture. In Figure 2 we compare the training time and corresponding memory usage for `VQ-TR` with the vanilla Transformer, as well as several Transformers that were specifically designed for computational efficiency, on the `Traffic` dataset.[9] For fairness, we used equal batch sizes and comparable hyperparameter values for all methods, as described in detail in the Appendix. We find that, when using a fairly moderate codebook size ($J = 25$), `VQ-TR` strongly outperforms all other methods in terms of memory efficiency, and is also competitive in terms of runtime. We also note that although the training time of `VQ-TR` was not the lowest in this comparison using a fixed batch size, in practice its substantially lower memory usage may allow for using larger batch sizes, and therefore faster training.

Finally, we provide more detailed results for VQ-TR, considering the effect of vector quantization, in Appendix C. There, we study the impact of the codebook size ($J$) on forecasting quality. Our main findings are that: (1) forecasting quality in general is not very sensitive to $J$; and (2) using small $J$ can have a positive regularizing effect, which is especially evident for `Traffic` and `Taxi` where the underlying signal is very simple and low dimensional.

---

[8]The full code will be published on acceptance, and hyperparameter details are provided in D.3.

[9]We also experimented with `Reformer`, but the training time and memory usage were so large (approx. 700s and 18GB respectively) that we excluded it from the comparison for clarity.

| Dataset | Method | CRPS | QL50 | QL90 | MSIS | NRMSE | sMAPE | MASE |
|---|---|---|---|---|---|---|---|---|
| Exchange | Trans-t | 0.018 | 0.022 | 0.014 | 56.26 | 0.035 | 0.030 | 4.834 |
| | Tft-t | 0.064 | 0.072 | 0.086 | 1647.64 | 0.087 | 0.328 | 55.77 |
| | Informer-t | 0.012 | 0.015 | 0.006 | 28.89 | 0.024 | 0.020 | 2.779 |
| | Autoformer-t | 0.014 | 0.019 | 0.006 | 19.16 | 0.027 | 0.022 | 3.591 |
| | ETSformer-t | 0.009 | 0.013 | 0.006 | **13.51** | 0.019 | 0.014 | 2.148 |
| | Hopfield-t | 0.016 | 0.018 | 0.012 | 46.46 | 0.031 | 0.027 | 4.208 |
| | Reformer-t | 0.018 | 0.022 | 0.007 | 95.09 | 0.031 | 0.027 | 6.044 |
| | Linformer-t | 0.014 | 0.018 | 0.008 | 37.98 | 0.026 | 0.020 | 2.822 |
| | Nystrom-t | 0.060 | 0.071 | 0.061 | 171.27 | 0.109 | 0.069 | 11.75 |
| | Longformer-t | 0.021 | 0.025 | 0.009 | 57.34 | 0.044 | 0.028 | 3.810 |
| | Performer-t | 0.063 | 0.070 | 0.018 | 206.4 | 0.092 | 0.066 | 8.963 |
| | PatchTST-t | 0.009 | 0.011 | 0.006 | 23.20 | 0.017 | **0.013** | **2.051** |
| | **VQ-TR-t** | **0.008** | **0.010** | **0.005** | 34.38 | **0.015** | 0.019 | 2.936 |
| Solar | Trans-t | 0.492 | 0.638 | 0.345 | 7.16 | 1.233 | 1.478 | 1.499 |
| | Tft-t | 0.931 | 0.995 | 1.305 | 48.04 | 2.03 | 1.950 | 1.950 |
| | Informer-t | 0.406 | 0.535 | 0.192 | 5.704 | 1.088 | 1.381 | 1.254 |
| | Autoformer-t | 0.758 | 0.985 | 0.308 | 15.68 | 2.035 | 1.854 | 2.317 |
| | ETSformer-t | 0.364 | 0.497 | 0.170 | 6.09 | 0.963 | 1.371 | 1.166 |
| | Hopfield-t | 0.477 | 0.642 | 0.243 | 5.94 | 1.217 | 1.471 | 1.505 |
| | Linformer-t | 0.984 | 1.083 | 1.270 | 45.46 | 1.933 | 1.776 | 2.556 |
| | Nystrom-t | 0.578 | 0.707 | 0.503 | 14.04 | 1.444 | 1.529 | 1.661 |
| | Longformer-t | 0.432 | 0.560 | 0.211 | 6.41 | 1.122 | 1.411 | 1.314 |
| | Performer-t | 0.472 | 0.626 | 0.294 | 6.29 | 1.205 | 1.466 | 1.474 |
| | PatchTST-t | 0.436 | 0.580 | 0.253 | 6.87 | 1.128 | 1.411 | 1.361 |
| | **VQ-TR-iqn** | **0.317** | **0.435** | **0.153** | **4.60** | **0.909** | **1.346** | **1.021** |
| Electricity | Trans-t | 0.061 | 0.078 | 0.035 | 7.49 | 0.538 | 0.115 | 0.853 |
| | Tft-t | **0.047** | **0.059** | **0.031** | **5.92** | 0.516 | **0.098** | **0.676** |
| | Informer-t | 0.064 | 0.079 | 0.054 | 6.47 | 0.739 | 0.116 | 0.788 |
| | Autoformer-t | 0.070 | 0.087 | 0.054 | 8.02 | 0.819 | 0.127 | 1.00 |
| | ETSformer-t | 0.068 | 0.081 | 0.064 | 8.43 | 0.650 | 0.128 | 0.904 |
| | Hopfield-t | 0.056 | 0.069 | 0.038 | 5.87 | 0.713 | 0.110 | 0.736 |
| | Reformer-t | 0.065 | 0.080 | 0.045 | 7.36 | 0.699 | 0.116 | 0.835 |
| | Linformer-t | 0.062 | 0.078 | 0.042 | 8.50 | 0.556 | 0.127 | 1.024 |
| | Longformer-t | 0.274 | 0.366 | 0.143 | 17.27 | 2.765 | 0.352 | 3.465 |
| | Performer-t | 0.163 | 0.202 | 0.119 | 20.20 | 1.326 | 0.248 | 2.470 |
| | PatchTST-t | 0.056 | 0.071 | 0.036 | 6.57 | 0.570 | 0.117 | 0.810 |
| | **VQ-TR-t** | 0.050 | 0.063 | 0.033 | 6.29 | **0.495** | 0.104 | 0.744 |
| Traffic | Trans-t | 0.241 | 0.294 | 0.172 | 11.50 | 0.521 | 0.394 | 1.300 |
| | TFT-t | 0.139 | 0.165 | 0.108 | 7.82 | 0.425 | 0.213 | 0.648 |
| | Informer-t | 0.117 | 0.138 | 0.096 | 6.81 | 0.404 | 0.148 | 0.528 |
| | Autoformer-t | 0.184 | 0.225 | 0.146 | 9.33 | 0.500 | 0.272 | 0.901 |
| | ETSformer-t | 0.165 | 0.197 | 0.137 | 9.35 | 0.495 | 0.260 | 0.783 |
| | Hopfield-t | 0.118 | 0.140 | 0.095 | **6.68** | 0.406 | 0.142 | 0.534 |
| | Linformer-t | 0.459 | 0.573 | 0.346 | 19.73 | 0.859 | 0.606 | 2.419 |
| | Nystrom-t | 0.272 | 0.336 | 0.175 | 13.07 | 0.555 | 0.387 | 1.528 |
| | Longformer-t | 0.317 | 0.382 | 0.278 | 15.92 | 0.694 | 0.556 | 1.651 |
| | Performer-t | 0.332 | 0.402 | 0.204 | 15.43 | 0.644 | 0.483 | 1.736 |
| | PatchTST-t | 0.166 | 0.209 | 0.130 | 8.16 | 0.506 | 0.211 | 0.812 |
| | **VQ-TR-t** | **0.110** | **0.130** | **0.093** | 6.91 | **0.392** | **0.137** | **0.500** |
| Taxi | Trans-nb | 0.308 | 0.388 | 0.212 | 6.09 | 0.628 | 0.594 | 0.790 |
| | Tft-nb | 0.301 | 0.377 | 0.211 | 6.27 | 0.617 | 0.584 | 0.767 |
| | Informer-nb | 0.326 | 0.407 | 0.230 | 7.11 | 0.649 | 0.634 | 0.825 |
| | Autoformer-nb | 0.365 | 0.458 | 0.273 | 7.38 | 0.726 | 0.648 | 0.916 |
| | ETSformer-nb | 0.311 | 0.393 | 0.211 | 5.85 | 0.634 | 0.597 | 0.797 |
| | Hopfield-nb | 0.340 | 0.424 | 0.265 | 6.91 | 0.685 | 0.634 | 0.850 |
| | Linformer-t | 0.648 | 0.951 | 0.493 | 8.32 | 1.094 | 1.804 | 1.855 |
| | Nystrom-t | 0.398 | 0.467 | 0.333 | 7.60 | 0.656 | 0.881 | 0.930 |
| | Longformer-nb | 0.398 | 0.473 | 0.320 | 7.40 | 0.652 | 0.905 | 0.937 |
| | Performer-nb | 0.397 | 0.471 | 0.297 | 7.18 | 0.626 | 0.954 | 0.954 |
| | PatchTST-nb | 0.301 | 0.386 | 0.203 | **5.03** | 0.611 | 0.599 | 0.785 |
| | **VQ-TR-nb** | **0.281** | **0.357** | **0.184** | 5.19 | **0.570** | **0.561** | **0.729** |
| Wikipedia | Trans-nb | 0.366 | 0.394 | 0.517 | 84.20 | 25.225 | 0.354 | 1.837 |
| | Tft-nb | 0.341 | 0.361 | 0.494 | 32.36 | 7.18 | 0.286 | 1.566 |
| | Informer-nb | 0.253 | 0.292 | 0.283 | 24.03 | 2.151 | 0.238 | 1.357 |
| | Hopfield-nb | 0.318 | 0.222 | 0.950 | 32.72 | 2.304 | 0.330 | 1.698 |
| | Longformer-nb | 0.529 | 0.487 | 0.677 | 59.04 | 2.571 | 0.479 | 2.211 |
| | Performer-nb | 0.461 | 0.401 | 0.491 | 31.68 | 2.283 | 0.326 | 1.762 |
| | PatchTST-nb | 0.256 | 0.305 | 0.272 | **20.20** | 2.101 | 0.249 | 1.507 |
| | **VQ-TR-iqn** | **0.231** | **0.269** | **0.260** | 21.17 | **2.121** | **0.213** | **1.269** |

Table 1: Forecasting metrics (lower is better) using Vanilla `Transformer` and other Transformer based models with Student-T (`-t`), Negative Binomial (`-nb`) or IQN (`-iqn`) emission heads, on the open datasets. The best (smallest) metrics are highlighted in **bold**. Note that the numbers for TFT are different than those in Lim et al. (2021a), since our `Traffic` dataset is different than theirs.

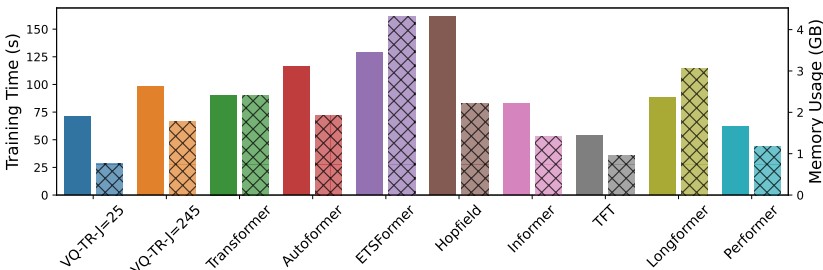

Figure 2: Training time and memory usage of `VQ-TR` compared with other transformer-based models on `Traffic` dataset, using comparable hyperparameters for all methods. For each method, the left bar displays training time (in seconds), and the right bar displays the memory usage (in gigabytes). In the case of VQ-TR we show results with a moderate codebook size ($J = 25$) as well as a very large codebook size ($J = 245$).

## 5 RELATED WORK

Our method decouples the input sequence from the computation of the attention block by utilizing a discrete set of latent. This strategy of reducing the computational cost is similar to the `Perceiver` (Hawthorne et al., 2022), `Set Transformer` (Lee et al., 2019), `Luna` (Ma et al., 2021), and `Compressive Transformer` (Rae et al., 2020) models. `Perceiver-AR` is the closest related method. However, it is a decoder-only architecture. Thus, to produce multiple samples for the probabilistic forecasting use case at inference time, `Perceiver-AR` requires running cross-attention over a large context window for $P$ times, causing both memory and computation bottlenecks. The use of VQ in sequential generative modeling has been explored in Audio/Speech settings (Dhariwal et al., 2020; Zeghidour et al., 2021; Baevski et al., 2020) where typically a VQ-VAE is trained on the data, and then a generative model is trained on these learned latent representations separately.

An alternative approach to improving the scalability of Transformer architectures to longer context windows is to better optimize the underlying computation, for example, using flash attention (Dao et al., 2022). Our work is orthogonal to such approaches, as it focuses on the scalability of the transformer architecture itself, and can further benefit from more optimized computation.

Using VQ for time-series forecasting problems has been explored by the `VQ-AR` (Rasul et al., 2022) model. This model works by applying an RNN to the raw time series input, and then applying VQ. This means, if one were to simply replace the RNN in `VQ-AR` with Transformer blocks, the transformer would be applied to non-quantized vectors, and we would still have quadratic-in-sequence-length scaling. In contrast, our work incorporates VQ *within the transformer architecture* as part of the encoder attention blocks. In Rabanser et al. (2020) the authors investigate the performance of forecasting models when they discretize the input into discrete bins whereas here we explicitly learn discrete representations.

## 6 SUMMARY AND DISCUSSION

We have presented `VQ-TR`, a novel transformer architecture that scales linearly with the encoder sequence size, and demonstrated its empirical value for probabilistic time series forecasting in a systematic comparison with other transformer-based methods for this task. We find that `VQ-TR` performs very competitively in terms of both forecasting performance and computation / memory usage, due to the dual efficiency and regularization benefits of vector quantization. Furthermore, our extensive comparison of the performance of state-of-the-art transformer methods for probabilistic time series forecasting can serve as a needed and previously missing benchmark within the literature. For future work, we would like to investigate the performance of `VQ-TR` for NLP, Audio, or Vision based problems.

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

## A  PROOF OF THEOREM 1

**Theorem A.1.** *If $\sum_u \left| \mathbf{q}_t^T \mathbf{k}_u - \hat{\mathbf{q}}_t^T \mathbf{k}_u \right| \leq \delta_t$ for sufficiently small $\delta_t > 0$, then the attention weights with respect to the approximation $\hat{\mathbf{q}}_t$, which we denote $\hat{w}_{tu}$, satisfy the bounds*

$$w_{tu}(1 - 2\delta_t) \leq \hat{w}_{tu} \leq w_{tu}(1 + 2\delta_t) \qquad \forall u \,.$$

*Further, as a result $|\mathbf{o}_t - \hat{\mathbf{o}}_t| \leq 2\delta_t \mathbf{o}_t$ holds element-wise for the output representation.*

*Proof.*

$$
\begin{aligned}
w_{tu} &= \frac{\exp(\mathbf{q}_t^T \mathbf{k}_u)}{\sum_j \exp(\mathbf{q}_t^T \mathbf{k}_j)} \\
&= \frac{\exp(\mathbf{q}_t^T \mathbf{k}_u - \hat{\mathbf{q}}_t^T \mathbf{k}_u + \hat{\mathbf{q}}_t^T \mathbf{k}_u)}{\sum_j \exp(\mathbf{q}_t^T \mathbf{k}_j - \hat{\mathbf{q}}_t^T \mathbf{k}_j + \hat{\mathbf{q}}_t^T \mathbf{k}_j)} \\
&= \frac{\exp(\hat{\mathbf{q}}_t^T \mathbf{k}_u) \exp(\mathbf{q}_t^T \mathbf{k}_u - \hat{\mathbf{q}}_t^T \mathbf{k}_u)}{\sum_j \exp(\hat{\mathbf{q}}_t^T \mathbf{k}_j) \exp(\mathbf{q}_t^T \mathbf{k}_j - \hat{\mathbf{q}}_t^T \mathbf{k}_j)} \,.
\end{aligned}
$$

We have $\sum_u \left| \mathbf{q}_t^T \mathbf{k}_u - \hat{\mathbf{q}}_t^T \mathbf{k}_u \right| \leq \delta_t$, and this directly implies $\max_u \left| \mathbf{q}_t^T \mathbf{k}_u - \hat{\mathbf{q}}_t^T \mathbf{k}_u \right| \leq \delta_t$.

Since, $\max_j \left| \mathbf{q}_t^T \mathbf{k}_j - \hat{\mathbf{q}}_t^T \mathbf{k}_j \right| \leq \delta_t$, then $\exp(-\delta_t) \leq \exp(\mathbf{q}_t^T \mathbf{k}_j - \hat{\mathbf{q}}_t^T \mathbf{k}_j) \leq \exp(\delta_t) \ \forall j$ and thus we have:

$$\exp(-2\delta_t) \leq \frac{w_{tu}}{\hat{w}_{tu}} \leq \exp(2\delta_t).$$

Assuming $\delta_t$ is small, $w_{tu}(1 - 2\delta_t) \leq \hat{w}_{tu} \leq w_{tu}(1 + 2\delta_t)$ or $|\hat{w}_{tu} - w_{tu}| \leq 2\delta_t w_{tu}$. Since $\mathbf{o}_t = \sum_u w_{tu} \mathbf{v}_u$,

$$|\mathbf{o}_t - \hat{\mathbf{o}}_t| \preceq \left| \sum_u w_{tu} \mathbf{v}_u - \hat{w}_{tu} \mathbf{v}_u \right| \preceq \sum_u |w_{tu} - \hat{w}_{tu}| \, \mathbf{v}_u \preceq \sum_u 2\delta_t w_{tu} \mathbf{v}_u \preceq 2\delta_t \mathbf{o}_t.$$

Here, $\preceq$ indicates element-wise inequality. $\qquad\square$

## B  ADDITIONAL RESULTS AND DISCUSSION

In Table 2 we provide the remaining results comparing `VQ-TR` against the non-transformer-based baselines. As noted in the main text, we mostly observe the same trend as with the comparison against transformer-based models, with `VQ-TR` performing best on most datasets/metrics, and close to best on most others. Specifically, on `Electricity`, `Traffic`, and `Taxi`, `VQ-TR` clearly outperforms the other methods, and on `Solar` and `Wikipedia`, `VQ-TR` is roughly on par with the best performing methods, with a tiny difference in each metric versus the best performing method.

However, the performance on `Exchange` requires some additional context and discussion. While `VQ-TR` mostly outperformed the other transformer-based methodologies on this dataset, it is slightly outperformed by some of the most simple non-Transformer baselines. A simple explanation for this is that `Exchange` consists of currency exchange rate time series, which are more less random walks with extremely little predictive information. Therefore, it is difficult to outperform simple classical methods like `ETS` on this kind of dataset. Despite this, we do believe it is notable that, although Transformer-based approaches do not seem to be ideal for datasets like `Exchange`, VQ-TR still outperforms the other Transformer-based methods here, which is additional evidence for the regularizing benefit of vector quantization.

## C  ABLATION STUDY ON VECTOR QUANTIZATION

Given our hypothesis on the possible regularizing benefit of vector quantization, we provide an ablation study on the effect of vector quantization on forecasting performance. Specifically, we experiment with varying the codebook size ($J$) in the range of $\{1, 2, 4, 8, 12, 16, 32, 50, 64, 100, 128, 245\}$, and for each value of $J$ we train `VQ-TR` with this codebook size, and record the CRPS metric on the

| Dataset | Method | CRPS | QL50 | QL90 | MSIS | NRMSE | sMAPE | MASE |
|---|---|---|---|---|---|---|---|---|
| Exchange | SQF-RNN-50 | 0.010 | 0.013 | 0.006 | **14.15** | 0.020 | 0.013 | 1.800 |
| | DeepAR-t | 0.012 | 0.016 | 0.007 | 69.29 | 0.022 | 0.030 | 9.980 |
| | ETS | 0.008 | **0.010** | 0.005 | 15.89 | 0.015 | **0.011** | **1.517** |
| | IQN-RNN | **0.007** | **0.010** | **0.004** | 17.37 | 0.014 | 0.013 | 3.041 |
| | MQCNN | 0.015 | 0.016 | 0.011 | 60.04 | 0.026 | 0.045 | 5.440 |
| | VQ-AR-t | 0.010 | 0.013 | 0.007 | 18.10 | 0.019 | 0.015 | 2.658 |
| | D-Linear-t | 0.010 | 0.013 | 0.006 | 24.69 | 0.020 | 0.015 | 2.506 |
| | **VQ-TR-t** | 0.008 | **0.010** | 0.005 | 34.38 | 0.015 | 0.019 | 2.936 |
| Solar | SQF-RNN-50 | 0.330 | 0.431 | 0.175 | 5.65 | 0.929 | **1.342** | 1.004 |
| | DeepAR-t | 0.418 | 0.543 | 0.254 | 7.33 | 1.072 | 1.393 | 1.275 |
| | ETS | 0.646 | 0.661 | 0.383 | 18.55 | 1.112 | 1.546 | 1.938 |
| | IQN-RNN | 0.373 | 0.491 | 0.165 | 5.99 | 1.037 | 1.356 | 1.150 |
| | MQCNN | 0.928 | 0.960 | 1.535 | 73.58 | 1.920 | 1.838 | 2.248 |
| | VQ-AR-iqn | 0.320 | **0.414** | 0.174 | 5.64 | **0.885** | 1.346 | **0.969** |
| | D-Linear-t | 0.451 | 0.545 | 0.338 | 13.24 | 1.079 | 1.391 | 1.280 |
| | **VQ-TR-iqn** | **0.317** | 0.435 | **0.153** | **4.60** | 0.909 | 1.346 | 1.021 |
| Electricity | SQF-RNN-50 | 0.078 | 0.097 | 0.044 | 8.66 | 0.632 | 0.144 | 1.051 |
| | DeepAR-t | 0.062 | 0.078 | 0.046 | 6.79 | 0.687 | 0.117 | 0.849 |
| | ETS | 0.076 | 0.100 | 0.050 | 9.99 | 0.838 | 0.156 | 1.247 |
| | IQN-RNN | 0.060 | 0.074 | 0.040 | 8.74 | 0.543 | 0.138 | 0.897 |
| | MQCNN | 0.129 | 0.148 | 0.132 | 30.54 | 1.230 | 0.240 | 2.000 |
| | VQ-AR-t | 0.054 | 0.068 | 0.036 | **5.88** | 0.653 | 0.107 | **0.717** |
| | D-Linear-t | 0.057 | 0.069 | 0.040 | 9.29 | 0.537 | 0.118 | 0.806 |
| | **VQ-TR-t** | **0.050** | **0.063** | **0.033** | 6.29 | **0.495** | **0.104** | 0.744 |
| Traffic | SQF-RNN-50 | 0.153 | 0.186 | 0.117 | 8.40 | 0.401 | 0.243 | 0.760 |
| | DeepAR-t | 0.172 | 0.216 | 0.117 | 8.02 | 0.472 | 0.244 | 0.890 |
| | ETS | 0.373 | 0.386 | 0.287 | 17.67 | 0.647 | 0.489 | 1.543 |
| | IQN-RNN | 0.139 | 0.168 | 0.117 | 7.11 | 0.433 | 0.171 | 0.656 |
| | MQCNN | 1.220 | 0.563 | 2.005 | 116.69 | 0.723 | 0.636 | 2.712 |
| | VQ-AR-t | 0.138 | 0.164 | 0.113 | 7.79 | 0.409 | 0.185 | 0.641 |
| | D-Linear-t | 0.182 | 0.216 | 0.139 | 9.25 | 0.481 | 0.221 | 0.851 |
| | **VQ-TR-t** | **0.110** | **0.130** | **0.093** | **6.91** | **0.392** | **0.137** | **0.500** |
| Taxi | SQF-RNN-50 | 0.286 | 0.362 | 0.188 | 5.53 | **0.570** | 0.609 | 0.741 |
| | DeepAR-nb | 0.299 | 0.379 | 0.203 | 5.44 | 0.610 | 0.582 | 0.771 |
| | ETS | 1.059 | 1.297 | 0.617 | 12.24 | 2.147 | 1.159 | 1.552 |
| | IQN-RNN | 0.295 | 0.370 | 0.201 | 6.51 | 0.583 | 0.629 | 0.758 |
| | MQCNN | 1.262 | 1.451 | 0.488 | 48.61 | 2.645 | 0.912 | 3.041 |
| | VQ-AR-nb | 0.286 | 0.362 | 0.193 | 5.43 | 0.572 | 0.570 | 0.741 |
| | D-Linear-nb | 0.335 | 0.422 | 0.236 | 6.02 | 0.678 | 0.641 | 0.854 |
| | **VQ-TR-nb** | **0.281** | **0.357** | **0.184** | **5.19** | **0.570** | **0.561** | **0.729** |
| Wikipedia | SQF-RNN-50 | 0.283 | 0.328 | 0.321 | 23.71 | 2.24 | 0.261 | 1.440 |
| | DeepAR-nb | 0.321 | 0.383 | 0.361 | 26.48 | 2.354 | 0.327 | 1.852 |
| | DeepAR-t | 0.235 | 0.27 | 0.267 | 23.77 | 2.15 | 0.219 | 1.295 |
| | ETS | 0.788 | 0.440 | 0.836 | 61.68 | 3.261 | 0.301 | 2.214 |
| | IQN-RNN | **0.221** | **0.254** | **0.251** | 21.78 | **2.102** | **0.193** | **1.214** |
| | MQCNN | 0.398 | 0.453 | 0.327 | 38.79 | 2.202 | 0.379 | 2.336 |
| | VQ-AR-iqn | 0.231 | 0.266 | 0.252 | 22.09 | 2.106 | 0.208 | 1.261 |
| | D-Linear-nb | 0.327 | 0.331 | 0.369 | 26.37 | 2.242 | 0.240 | 1.430 |
| | **VQ-TR-iqn** | 0.231 | 0.269 | 0.260 | **21.17** | 2.121 | 0.213 | 1.269 |

Table 2: Forecasting metrics (lower is better) using: `SQF-RNN` with 50 knots, `ETS`, `MQCNN`, and `IQN-RNN`, `DeepAR`, `VQ-AR` and **`VQ-TR`** with Student-T (`-t`), Negative Binomial (`-nb`) or IQN (`-iqn`) emission heads, on the open datasets. The best (smallest) metrics are highlighted in bold.

test set. We performed this experiment on all of our datasets, and for each repeated it 5 times, with hyperparameters used detailed in the Appendix.

We summarize the results of this ablation study in Figure 3. In general, we typically see good performance with relatively small $J$, with performance often degrading when $J$ is large. This seems to especially hold for both the `Taxi` and `Traffic` datasets, which is notable because each of these datasets exhibits very simple low-dimensional underlying dynamics despite having large numbers of time series, with fairly simple shared day/night cyclical behavior. In other words, for the exact datasets where, given domain knowledge, we would expect regularization to be most important, we indeed see that `VQ-TR` benefits significantly from smaller codebook sizes. Therefore, the `VQ-TR` architecture can provide benefits for forecasting performance as well as computational efficiency.

Note that the scale of variation seen in these plots is very small compared with the differences in CRPS values observed for different methods in Table 2. For example, with the `Taxi` dataset we see CRPS ranging from approx. 0.29 to 0.305 in Figure 3; even the most pessimistic end of this range is barely any worse than the most competitive benchmark methods (CRPS of 0.301), and this is despite the fact that the comparison favors the benchmarks since we are comparing `VQ-TR` performance without any dataset-specific hyperparameter optimization (other than codebook size) to benchmark performance with hyperparameter optimization (as seen in Table 2, when we do full hyperparameter

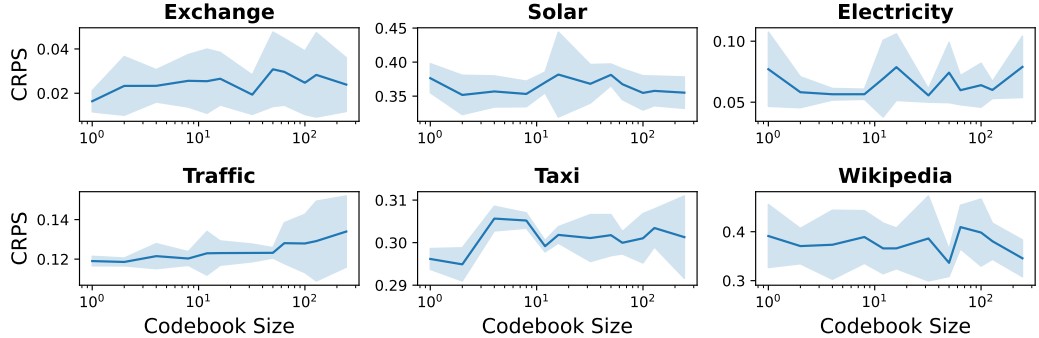

Figure 3: Effect of varying the codebook size ($J$) on the test set CRPS metric. Lines are mean CRPS values, and shaded regions represent standard deviation over 5 runs.

optimization for `VQ-TR` on `Taxi` we get CRPS of $0.281$). Similar results also hold for the other datasets. In other words, although we see some interesting trends in Figure 3, on the whole `VQ-TR` is not very sensitive to the codebook size, and performs competitively even when codebook sizes are not well optimized.

Finally, an interesting observation is that we can obtain decent performance even with $J = 1$, since a single vector $\mathbf{z}_1$ will get *cross-attended* with respect to the variable inputs, and then fed into the vanilla Transformer decoder. This is unlike `VQ-AR` (Rasul et al., 2022), where $J = 1$ does not work.

## D  ADDITIONAL EXPERIMENT DETAILS

### D.1  DATASET DETAILS

We summarize the details of the datasets used in our experiments in Table 3.

| Dataset | $D$ | Dom. | Freq. | Time step | Pred. len. |
|---|---|---|---|---|---|
| Exchange | 8 | $\mathbb{R}^{\geq 0}$ | day | $6,071$ | 30 |
| Solar | 137 | $\mathbb{R}^{\geq 0}$ | hour | $7,009$ | 24 |
| Electricity | 321 | $\mathbb{R}^{\geq 0}$ | hour | $15,782$ | 48 |
| Traffic | 862 | $(0, 1)$ | hour | $14,036$ | 24 |
| Taxi | $1,214$ | $\mathbb{N}^{\geq 0}$ | 30-min | $1,488$ | 24 |
| Wikipedia | $9,535$ | $\mathbb{N}^{\geq 0}$ | day | 762 | 30 |

Table 3: Dataset statistics. For each dataset, we list: ($D$) the number of time series; (Dom.) the domain of time series values; (Freq.) the frequency of the time series; (Time step) the total number of time steps for the training data; and (Pred. len.) the prediction length used.

### D.2  DETAILS OF BASELINES

In our experiments, we compared `VQ-TR` against the following baseline models:

- `DeepAR` (Salinas et al., 2019b): an RNN based probabilistic model which learns the parameters of some chosen distribution for the next time point;

- `MQCNN` (Wen et al., 2017): a Convolutional Neural Network model which outputs chosen quantiles of the forecast upon which we regress the ground truth via Quantile loss;

- `SQF-RNN` (Gasthaus et al., 2019): an RNN based non-parametric method which models the quantiles via linear splines and also regresses the Quantile loss;

- `IQN-RNN` (Gouttes et al., 2021): combines an RNN model with an Implicit Quantile Network (IQN) (Dabney et al., 2018) head to learn the distribution similar to `SQF-RNN`;

- `VQ-AR` (Rasul et al., 2022): an RNN based encoder-decoder model which quantizes its input via a VQ;

- `D-Linear` (Zeng et al., 2023): a linear model which decomposes the time series into trend and its residual;

- `ETS` (Hyndman and Khandakar, 2008): exponential smoothing method using weighted averages of past observations with exponentially decaying weights as the observations get older together with Gaussian *additive* errors (E) modeling trend (T) and seasonality (S) effects separately

- `TFT` (Lim et al., 2021b): an auto-regressive attention based Seq-to-Seq model with variable selection network for selecting relevant inputs;

- `Informer` (Zhou et al., 2021): an efficient transformer and full horizon predictor model;

- `Autoformer` (Wu et al., 2021): a transformer that decomposes the trend and seasonal components during the forecasting process together with a series-wise auto-correlation mechanism;

- `ETSformer` (Woo et al., 2022): a transformer architecture that adds the principle of exponential smoothing and frequency attention in the attention mechanism;

- `Hopfield` (Ramsauer et al., 2021): a modern Hopfield network with continuous state which generalizes attention;

- `Longformer` (Beltagy et al., 2020): a local attention model with sliding window attention;

- `Reformer` (Kitaev et al., 2020): a transformer that replaces dot-product attention by locality-sensitive hashing and reversible residual layers which reduces the memory footprint in the backward pass;

- `Linformer` (Wang et al., 2020): a transformer that learns a fixed projection matrix to reduce the length of the keys and value;

- `Nystromformer` (Xiong et al., 2021): a transformer that adapts the Nyström method to approximate standard self-attention to make its complexity linear;

- `Performer` (Choromanski et al., 2021): a transformer model which estimates regular full-rank attention by using linear space/compute complexity;

- `PatchTST` (Nie et al., 2023): a transformer encoder that takes as input segments of time series as fixed-sized vectors;

### D.3 TRAINING AND HYPERPARAMETER DETAILS FOR MAIN EXPERIMENTS

All the models have been trained using the hyperparameters from their respective papers with Student-T (`-t`), Negative Binomial (`-nb`), or Implicit Quantile Network (`-iqn`) emission heads. Note that we can afford to use a longer context length of $C = 20 \times P$ for `VQ-TR` due to its memory efficiency, as noted in Figure 2. Here, $P$ is the prediction horizon for each dataset. We use two encoder layers and six decoder layers, i.e., $N = 2$ and $M = 6$. We use $J = 25$ codebook vectors and train with a batch size of 256 for 20 epochs using the Adam (Kingma and Ba, 2015) optimizer with default parameters and a learning rate of 0.001. At inference time, we sample $S = 100$ times for each time point and feed these samples in *parallel* via the batch dimension autoregressively through the decoder to produce the reported metrics. Full complete details for running these experiments will be available with the code release.

### D.4 ADDITIONAL DETAILS FOR COMPUTATIONAL EFFICIENCY COMPARISON

For our computational efficiency comparison summarized in Figure 2, we compared all methods by training them over 20 epochs, with a batch size of 128, with training details otherwise the same as described for our main experiments. As described in the main text, we compared all methods using hyperparameters that were as similar as possible. Specifically, for each method, where possible, we used 2 encoding layers and 2 decoding layers (*i.e.* $N = M = 2$). For `VQ-TR` we set $J = 25$. Other hyperparameters for all methods were hand-tuned so that all methods had as close to the same

number of trainable hyperparameters as `VQ-TR` as possible. The experiments were performed on a single Tesla V100S GPU with 32GB of RAM. In all cases, we computed runtimes by performing this experiment 5 times and report the median runtime; however, we note that the variance in runtimes for all methods was extremely negligible (of order $\approx 1$s). Full complete details for performing this comparison will be available with the code release.

## D.5   ADDITIONAL DETAILS FOR ABLATION STUDY ON VECTOR QUANTIZATION

For this ablation study, we used the same training details and other hyperparameter values for every codebook size $J \in \{1, 2, 4, 8, 12, 16, 32, 50, 64, 100, 128, 245\}$. Specifically, we used 20 epochs with a batch size of 256, and with other training details the same as for the main experiment. For the `VQ-TR` architecture we used one encoding layer and four decoding layers, *i.e.*, $N = 1$ and $M = 4$. Again, full complete details for performing this ablation study will be available with the code release.

## D.6   `VQ-TR` IMPLEMENTATION DETAILS

We provide the code for the `VQ-TR` encoder below. In addition, as noted in the main text, full code for all methods shall be made available upon acceptance.

```python
import torch
import torch.nn as nn
import torch.nn.functional as F
from torch import einsum

# https://github.com/lucidrains/vector-quantize-pytorch
from vector_quantize_pytorch import VectorQuantize

def FeedForward(dim, hidden_dim, dropout=0.0):
    return nn.Sequential(
        nn.LayerNorm(dim),
        nn.Linear(dim, hidden_dim, bias=False),
        nn.GELU(),
        nn.Dropout(dropout),
        nn.Linear(hidden_dim, dim, bias=False),
    )

class Attention(nn.Module):
    def __init__(self, dim, dim_head=64, heads=8, dropout=0.0):
        super().__init__()
        self.scale = dim_head**-0.5
        self.heads = heads
        inner_dim = heads * dim_head

        self.norm = nn.LayerNorm(dim)
        self.dropout = nn.Dropout(dropout)
        self.to_qkv = nn.Linear(dim, inner_dim * 3, bias=False)
        self.to_out = nn.Linear(inner_dim, dim, bias=False)

    def forward(self, x):
        x = self.norm(x)

        q, k, v = self.to_qkv(x).chunk(3, dim=-1)
        q, k, v = map(
            lambda t: rearrange(t, "b n (h d) -> b h n d", h=self.heads),
                                                (q, k, v)
        )

        q = q * self.scale

        sim = einsum("b h i d, b h j d -> b h i j", q, k)
```

```python
        attn = sim.softmax(dim=-1)
        attn = self.dropout(attn)

        out = einsum("b h i j, b h j d -> b h i d", attn, v)

        out = rearrange(out, "b h n d -> b n (h d)")
        return self.to_out(out)

class VQAttention(nn.Module):
    def __init__(
        self,
        dim,
        codebook_size,
        dim_feedforward=16,
        dim_head=16,
        heads=2,
        max_heads_process=2,
        dropout=0.0,
        cross_attn_dropout=0.0,
        depth=1,
        decay=0.8,
        commitment_weight=1.0,
    ):
        super().__init__()
        self.scale = dim_head**-0.5
        self.heads = heads
        self.max_heads_process = max_heads_process

        inner_dim = heads * dim_head
        self.dim = dim

        self.norm = nn.LayerNorm(dim)
        self.context_norm = nn.LayerNorm(dim)
        self.dropout = nn.Dropout(dropout)

        # drop out a percentage of the prefix during training,
        # shown to help prevent overfitting
        self.cross_attn_dropout = cross_attn_dropout

        self.to_q = nn.Linear(dim, inner_dim, bias=False)
        self.to_kv = nn.Linear(dim, inner_dim * 2, bias=False)
        self.to_out = nn.Linear(inner_dim, dim)

        self.vq = VectorQuantize(
            dim=dim,
            codebook_size=codebook_size,
            decay=decay,
            commitment_weight=commitment_weight,
            threshold_ema_dead_code=2,
        )

        self.layers = nn.ModuleList([])
        for _ in range(depth):
            self.layers.append(
                nn.ModuleList(
                    [
                        Attention(
                            dim=dim, dim_head=dim_head, heads=heads,
                                                        dropout=
                                                        dropout
                        ),
                        FeedForward(dim, hidden_dim=dim_feedforward,
                                                        dropout=
                                                        dropout),
```

```python
                ]
            )
        )

    def forward(self, context_input, context_mask=None):
        batch, context_len, device = (
            context_input.shape[0],
            context_input.shape[-2],
            context_input.device,
        )

        # take care of cross attention dropout
        if self.training and self.cross_attn_dropout > 0.0:
            rand = torch.zeros((batch, context_len), device=device).
                                                uniform_()
            keep_context_len = context_len - int(context_len * self.
                                                cross_attn_dropout)
            keep_indices = rand.topk(keep_context_len, dim=-1).indices
            keep_mask = torch.zeros_like(rand).scatter_(1, keep_indices,
                                                1).bool()

            context_input = rearrange(
                context_input[keep_mask], "(b n) d -> b n d", b=batch
            )

            if context_mask is not None:
                context_mask = rearrange(
                    context_mask[keep_mask], "(b n) -> b n", b=batch
                )

        _, indices, commit_loss = self.vq(context_input)

        x = repeat(self.vq.codebook, "m d -> b m d", b=batch)  # [B, M, D
                                                ]

        # normalization
        x = self.norm(x)
        context = self.context_norm(context_input)

        # derive queries, keys, values
        q = self.to_q(x)
        k, v = self.to_kv(context).chunk(2, dim=-1)

        q, k, v = map(
            lambda t: rearrange(t, "b n (h d) -> b h n d", h=self.heads),
                                                (q, k, v)
        )
        q = q * self.scale

        # take care of masking
        i, j = q.shape[-2], k.shape[-2]
        mask_value = -torch.finfo(q.dtype).max

        if context_mask is not None:
            mask_len = context_mask.shape[-1]
            context_mask = F.pad(context_mask, (0, max(j - mask_len, 0)),
                                                value=True)
            context_mask = rearrange(context_mask, "b j -> b 1 1 j")

        # process in chunks of heads
        out = []
        max_heads = self.max_heads_process
        for q_chunk, k_chunk, v_chunk in zip(
            q.split(max_heads, dim=1),
            k.split(max_heads, dim=1),
```

```python
            v.split(max_heads, dim=1),
        ):
            sim = einsum("b h i d, b h j d -> b h i j", q_chunk, k_chunk)

            if context_mask is not None:
                sim = sim.masked_fill(~context_mask, mask_value)

            attn = sim.softmax(dim=-1)
            attn = self.dropout(attn)

            out_chunk = einsum("b h i j, b h j d -> b h i d", attn,
                                               v_chunk)
            out.append(out_chunk)

        # concat all the heads together
        out = torch.cat(out, dim=1)

        # merge heads and then combine with linear
        out = rearrange(out, "b h n d -> b n (h d)")
        out = self.to_out(out)

        # self-attention on latents
        for attn, ff in self.layers:
            out = attn(out) + out
            out = ff(out) + out

        expanded_indices = indices.unsqueeze(-1).expand(*indices.shape,
                                         self.dim)
        outputs = torch.gather(out, 1, expanded_indices)

        return outputs, commit_loss

class VQTrModel(nn.Module):
    def __init__(self, ...):
        ...

        # VQ attention encoder
        self.encoder = nn.ModuleList([])
        for _ in range(num_encoder_layers):
            self.encoder.append(
                VQAttention(
                    codebook_size=codebook_size,
                    decay=decay,
                    commitment_weight=commitment_weight,
                    dim=d_model,
                    depth=depth,
                    heads=nhead,
                    dim_head=dim_head,
                    dim_feedforward=dim_feedforward,
                    dropout=dropout,
                    cross_attn_dropout=dropout,
                ),
            )

        # causal decoder and mask
        decoder_norm = nn.LayerNorm(d_model, eps=1e-5)
        decoder_layer = nn.TransformerDecoderLayer(
            d_model,
            nhead,
            dim_feedforward,
            dropout,
            activation,
            layer_norm_eps=1e-5,
            batch_first=True,
```

```
            norm_first=False,
        )
        self.decoder = nn.TransformerDecoder(
            decoder_layer, num_decoder_layers, decoder_norm
        )

        # causal decoder tgt mask
        self.register_buffer(
            "tgt_mask",
            nn.Transformer.generate_square_subsequent_mask(
                                              prediction_length),
        )
    ...
```

