# OpenReview forum: "VQ-TR: Vector Quantized Attention for Time Series Forecasting"
_ICLR.cc/2024/Conference — ICLR 2024 poster_

### Official Review · Reviewer_9deP · 2023-10-28

**Soundness:** 4 excellent
**Presentation:** 3 good
**Contribution:** 3 good
**Rating:** 6
**Confidence:** 4

**Summary:**

The manuscript proposes a transformer architecture for time series forecasting, that uses vector quantization to scale the encoder sequence size, resulting in faster training time, less memory usage, and time series forecasting results from multiple metrics.

**Strengths:**

To adopt vector quantization in Transformer based models for time series forecasting is novel to the best of my knowledge.
The manuscript is overall well and clearly written.
The proposed method has the potential to benefit the time series modelling community.

**Weaknesses:**

- From Fig. 3, the influence of the codebook size J to the VQ-TR seems not stable. A more detailed discussion on this might help potential users to select this hyperparameter when using the model.

 - The manuscript claims that
>...methods based on Transformers (Vaswani et al., 2017) have dominated the state-of-the-art, outperforming both classical autoregressive approaches, as well as deep learning approaches using Convolutional Neural Networks (CNNs) or Recurrent Neural Networks (RNNs).

   However, in e.g. [1], which is not a Transformer based model, results of CRPS are better than the ones reported in the manuscript.\
   A discussion (or comparison) on [1] might help to promote the contribution of the manuscript.

 - The manuscript assumes that the distribution of the predicted time point follows a (conditional) Gaussian distribution. Is there any theoretical or empirical evidence supporting this assumption? In e.g. [2], the paper also proposes a model for probabilistic time series forecasting, with a more solid assumption that the frequencies follow Gaussian distribution. A discussion on this might help to understand this Gaussian assumption.

 - In Sec. 4.2, the use of ```\citet{}``` and ```\citep{}``` needs to be carefully handled.




---
[1] Kashif Rasul et al. Multivariate Probabilistic Time Series Forecasting via Conditioned Normalizing Flows, ICLR 2021\
[2] Zhongjie Yu et al. Predictive Whittle networks for time series. UAI 2022

**Questions:**

- Is it true that the model as proposed in the manuscript, works for univariate time series only?

 - Can you elaborate the influence of different codebook sizes?

 - How is the context window length C selected?

---

> ### Author Response · Authors · 2023-11-16
> **Response to Reviewer 9deP**
>
> Thank you for taking the time to review our paper, for your insightful comments, and for acknowledging the novelty and benefit of our methodology! We will fix the formatting issues you noted regarding citations, and we provide detailed responses to your questions and criticisms below.
>
> # Influence of Codebook Size
>
> Please see our general comments to all reviewers (titled: “Response RE Codebook Sizes”) for a very detailed discussion about the influence of codebook size, and the stability of this effect. In summary, the main takeaway from the paper should be that for the most part performance is fairly robust to codebook size, with changes in mean performance due to varying codebook size that are very small in absolute value. However, we do see evidence that in some cases, especially for datasets where the underlying structure is simpler and strong regularization is needed (i.e. Traffic and Taxi), there is an empirical benefit of using especially small codebook sizes. We will add additional discussions to the final paper to make the influence of changing the codebook size more clear.
>
> # Assumption of Gaussian Model
>
> To clarify, we do *not* actually assume that the time series data is conditionally Gaussian. In section 2.1 we use conditional Gaussian distributions as an example of how we might define a concrete probabilistic model for time series that is parameterized by transformers, but as mentioned there this is only an example. In practice, we can use our VQ-TR methodology with any distribution class for the emission heads. In other words, the user specifies some parametric class of distributions (e.g. Gaussian, student-t, negative binomial, etc.), which is what we call the “emission head”, and the VQ-TR model predicts the distribution parameters (the shape and number of these parameters depends on the user’s chosen class) of the next data point. Since these predicted parameters index a class of probability distributions, this corresponds to predicting the distribution of the next data point. In practice, in our experiments, we use VQ-TR with either implicit quantile, student-t, or negative binomial emission heads; see our response to reviewer XX8d for our rationale of deciding which distribution was used on which dataset. In all cases, this is a modeling choice that of course won’t be exactly correct, but we believe these are flexible enough choices to learn good probabilistic models (as evidenced by our empirical results). Also, note that in no cases do we actually use Gaussian distributions with VQ-TR in our experiments. We will make this more clear in our final paper.
>
> # Reported Performance in Our Results versus in Literature
>
> In short, the CRPS metric you are most likely referring to in [1] is different from the CRPS metric we are computing in our paper; the former takes smaller values than the latter. This is a common misunderstanding in the literature, and we appreciate the opportunity to clarify.
>
> In more explicit detail, note that [1] is a multivariate model, that learns the full joint distribution of all time series together; the results you most likely refer to are in terms of the CRPS-SUM metric, which is different from the univariate metrics CRPS. Importantly for comparing these works, CRPS-SUM is smaller than the corresponding CRPS metric. The appendix of [1] contains the CRPS table where one observes a recurring pattern in time series research, namely that the univariate models (i.e. all those considered in our paper) typically perform better than their multivariate counterparts for the smallish datasets being used in scientific research. This phenomenon can also be observed in the DLinear or PatchTST and similar papers where even a simple linear model is shown to be better than transformer-based models, although this point of multivariate-vs-univariate is not made clear in those works either.
>
> # VQ-TR with Univariate versus Multivariate Time Series
>
> There is no reason why the VQ-TR architecture cannot be used for multivariate time series modeling. That being said, as discussed above, it is known that doing separate univariate time series modeling for each time series tends to outperform multivariate approaches for the kinds of smallish datasets used in scientific research, so therefore our implementation of VQ-TR used in our experiments is univariate (as with all baselines compared against.)
>
> # How is Context Length Chosen
>
> See our general response to all reviewers for detail about this (titled: “Response RE Context Window Sizes”). In short, the context window is a hyperparameter that is optimized on a per method/dataset basis.

---

> > ### Comment · Reviewer_9deP · 2023-11-20
> >
> > Thanks to the authors for the answer to my questions, and most of my questions are addressed.\
> > My remaining questions are:
> > 1) In my humble opinion, if conditional Gaussian is in the end not used, then using it as an example is a bit misleading.
> > 2) A more precise description of the multivariate time series question is, when the current version of VQ-TR is used for general multivariate time series, does it model each time series independently, or jointly?

---

> > > ### Author Response · Authors · 2023-11-21
> > >
> > > Thank you for the questions.
> > >
> > > 1. We apologize that you found our presentation using Gaussian distributions as an example confusing. We will make our presentation clearer with the following: (a) we will change the example in section 2.1 to be in terms of a student-t distribution head, rather than a Gaussian distribution head; and (b) we will explicitly note in section 2.1 that in practice we both can and do use different distribution heads for different datasets.
> > >
> > > 2. To be very explicit, for such multi time series problems VQ-TR models each time series independently, using a shared model. For example, if we have a multivariate dataset with K time series, the future values for each of those time series are assumed to depend only on past values of that specific time series, and not depend on any values of the other K-1 time series. However, we do use a common neural net model for modeling the relationship between past and future values, which is applied independently to each of the K time series. Also, note that this way of modeling the time series independently is used for all models in our comparison, not just VQ-TR. We will add a similarly explicit explanation to the final paper.

---

> > > > ### Comment · Reviewer_9deP · 2023-11-21
> > > >
> > > > Thank you for the follow-up answer. I am keeping my positive score for now and can increase the score when I see an updated revision that addressing the issues.

---

### Official Review · Reviewer_JnRg · 2023-10-31

**Soundness:** 3 good
**Presentation:** 3 good
**Contribution:** 2 fair
**Rating:** 6
**Confidence:** 4

**Summary:**

The paper addresses the challenges in probabilistic time series forecasting, which involve long sequences, and the need for a large number of samples for accurate inference. The paper highlights that current state-of-the-art methods based on Transformers are computationally inefficient due to their quadratic complexity in sequence length and primarily focus on non-probabilistic point estimation. To overcome these limitations, the paper introduces a novel transformer architecture called VQ-TR. VQ-TR utilizes a discrete set of latent representations in the Attention module, enabling linear scaling with the sequence length and providing effective regularization to prevent overfitting. The authors compare several Transformer-based time series forecasting methods for probabilistic forecasting, and VQ-TR demonstrates competitive performance in forecasting accuracy, computational efficiency, and memory usage.

**Strengths:**

The manuscript is well-written, with clear and easily understandable ideas. The paper offers a thorough evaluation, encompassing multiple baseline models across various benchmark datasets. It effectively illustrates the impact of the VQ module, not only in terms of performance enhancement but also in computational and memory efficiency.

**Weaknesses:**

The paper suffers from a significant lack of novelty, as its approach closely mirrors that of VQ-AR by Rasul et al. (2022). Essentially, the method merely replaces the RNN with a Transformer model. Given this redundant contribution, the paper falls short of meeting the standards expected for a conference like ICLR and might be better suited as a workshop paper.

Furthermore, in Figure 3, the analysis of how codebook size affects overall performance, fails to provide a clear conclusion or discernible trends regarding the impact of codebook size. It appears that determining the correct size is critical, and achieving the desired performance may require a careful HP tuning for a given dataset.

**Questions:**

How many attempts were done to find the right codebook size for the presented experiments? Adding the computing costs of tuning the codebook size, how well VQ-TR compares with methods that provides comparable performance, when trained using default settings?

---

> ### Author Response · Authors · 2023-11-16
> **Response to Reviewer JnRg**
>
> Thank you for taking the time to review our paper, for your thoughtful feedback, and for acknowledging our thoroughness in demonstrating the empirical benefit of VQ-TR. We provide detailed comments to your major criticisms and questions below.
>
> # Novelty of our Methodology
>
> We would like to very respectfully, but firmly, push back against the assertion that our methodology is only a minor variation on existing methods. We provide a detailed discussion about this in our general response to all reviewers  (titled: “Response RE VQ-TR vs VQ-AR”), but in summary, this assertion is not correct. As explained there, replacing the RNN in VQ-AR with a transformer would not give VQ-TR, and would not change the “quadratic in context length” transformer complexity. Rather, VQ-TR is a different methodology, based on a novel way of placing the VQ module inside the transformer (rather than on the transformer outputs, which is what a simple extension of VQ-AR to transformers would do). Given this, we strongly believe that our work contains strong methodological novelty appropriate for ML conference publication. However, we apologize that this novelty was as not clear as it could have been, and we will make this more clear in our final paper.
>
> # Impact of Codebook Size on Cost and Performance
>
> Regarding your points about the importance of codebook size on performance and the cost of codebook size optimization, we provide a very detailed discussion of this in our general response to all reviewers (titled: “Response RE Codebook Sizes”). To summarize the discussion there, although at first glance Figure 3 may make it look like VQ-TR is sensitive to the codebook size, it is really not. Both the observed mean effects of changing the codebook sizes, and the variance of performance, are very small in absolute terms, especially when compared with the difference in performance between methods. In other words, one of the key takeaways from Figure 3 should be that VQ-TR is *not* overly sensitive to codebook size.
>
> Despite this lack of sensitivity, however, we agree with you that if one wants to fully optimize performance, then it is relevant to consider the computational impact of codebook size. We provide some detail in our general response to all reviewers about the effect of codebook size on computational cost. In short, increasing codebook size does not increase the cost overly much, and even increasing it to very large values beyond what is needed for good performance results in a computational cost that is at worst on par with other methods. Since, as evidenced by Figure 3, we can always obtain (approximately) optimal performance with relatively small codebook sizes, our results in Figure 2 should be fairly reflective of the actual computational cost of codebook size optimization in practice. However, in order to be extra thorough, we will include in the appendix a similar plot as in Figure 2 for a wide range of codebook sizes, and we will add an extra discussion along these lines.
>
> Regarding the questions about how many hyperparameter configurations we tried for VQ-TR, and the cost of optimizing VQ-TR versus running baselines with default values, we note that for *all* methods in our main empirical evaluation we performed hyperparameter optimization over a wide range of values for all important hyperparameters on a per-dataset basis (which included the codebook size in the case of VQ-TR). While of course, it would be possible to compare the computational cost of VQ-TR with full hyperparameter optimization versus the computational costs of baselines with no hyperparameter optimization, this seems unfair and inappropriate, as this is not how their forecasting performance was compared. Although of course getting the most performance out of VQ-TR requires some hyperparameter tuning, this is also true for all other methods, and in all cases, the computational cost is inflated by the number of hyperparameter configurations used. Furthermore, as discussed above, and contrary to the premise of this question, VQ-TR is *not* very sensitive to the exact codebook size as long as it is within some range of values that seems generically reasonable for all datasets that we investigated (e.g. between 10 and 100).
>
> Finally, regarding your comment about the lack of clear trends in the codebook size comparison, while we agree that there aren’t extremely strong trends that are consistent across every dataset, we do see some interesting trends in some datasets. We provide a more detailed discussion about this in our response to all reviewers, and make the case more explicitly that the performance trend in the Taxi and Traffic datasets is interesting, and indeed evidence of a beneficial regularizing effect of VQ. We will make our explanation of this trend more clear in the final paper.

---

> > ### Comment · Reviewer_JnRg · 2023-11-22
> >
> > I appreciate the authors for addressing my concerns with detailed explanations. Although incorporating the VQ module directly into the transformer enhances computational efficiency, the proposed method and its motivations are still largely influenced by VQ-AR. Additionally, I agree with other reviewers that the paper could be interesting for the broader machine learning community, not just in time series forecasting. Therefore, I am changing my score to positive.

---

### Official Review · Reviewer_XX8d · 2023-10-31

**Soundness:** 4 excellent
**Presentation:** 4 excellent
**Contribution:** 3 good
**Rating:** 8
**Confidence:** 4

**Summary:**

This work introduces VQ-TR, combining vector quantization and the transformer architecture for time series forecasting. Specifically, vector quantization limits query vectors to be one of a fixed size codebook, which implies that any subsequent cross/self-attention computation scales with the codebook size instead of the context length. A masked attention decoder is used to 'unroll' the forecast. Apart from favorable computational properties, the method appears to clearly outperform the SoTA in transformer-based point and probabilistic forecasting.

**Strengths:**

The paper proposes a simple and highly practical tool for forecasting, and helps resolve an ongoing discussion in the forecasting community about the use of transformers by positing that a 'quantized' small-scale transformer already outperforms many of the current architectures in use.

The extensive experimental results posted are themselves of interest to the community and the work can potentially benefit other areas where transformers are often applied.

**Weaknesses:**

Motivation, exposition and the key idea follow closely from VQ-AR. This is however justified since the computational benefits to the transformer architecture are stronger. However, the contrast could be made clearer in the paper.

Also, key design decisions about experiments including number of layers, codebook size used to produce the tables are currently only available in the appendix. Please consider moving these important details forward.

Importantly, the experiment presentation is somewhat misleading. The main paper only compares against transformer baselines where the proposed method appears to categorically outperform the SoTA, however the non-transformer based baselines are moved to the appendix where results are mixed against non-transformer baselines; and less competitive than what the main paper Section 4.4 claims.

A last open point that should be addressed is the choice of the distribution output per dataset. How was this choice made for the baselines? The fact that -iqn and -t survive for VQ-TR in wiki and taxi respectively invites the question how these hyperparameters were selected.

**Questions:**

- Why does the model have higher runtime than TFT? Despite the comment that similar batch sizes were used, one would expect that the quantized architecture still outperforms TFT.
- In the appendix the authors write: "Note that we can afford to use a longer context length of C = 20 × P for VQ-TR due to its memory efficiency, as noted in Figure 2." Was the context length fixed across models or did VQ-TR get a head start?

---

> ### Author Response · Authors · 2023-11-16
> **Response to Reviewer XX8d**
>
> Thank you for taking the time to review our paper, and for your helpful comments. We especially appreciate you for recognizing the practical nature of our contribution, as well as the extensiveness of our comparison! Indeed, we believe that we have gone above and beyond to provide an empirical comparison of probabilistic time series forecasting performance for modern methodologies that is far more extensive than anything else we are familiar with in the literature. We provide specific responses to most of your criticisms and questions below. For our response to your comment about the closeness of our methodology with VQ-AR, and about what context windows were used in experiments, please see our general responses to all reviewers (titled: “Response RE VQ-TR vs VQ-AR” and “Response RE Context Window Sizes”).
>
> # Experimental Details in Appendix
>
> We appreciate your feedback about this issue. While we believe that it is common practice for ML conference papers to provide many of these kinds of details in the appendix, we agree that it would make the paper more readable to provide some of this info in the main paper, such as e.g. the codebook size used for our computational comparison in Figure 2. We will make some adjustments along these lines.
>
> # Presentation of Experiment Results
>
> While it is true that the results for VQ-TR seem slightly more mixed in Table 3 than Table 1, for the most part (with one notable exception described below) the same kinds of trends are present. In more detail:
>
> 1. On Electricity, Traffic, and Taxi, VQ-TR clearly outperforms other methods
>
> 2. On Solar and Wikipedia, VQ-TR is roughly on par with the best performing methods, with a tiny difference in each metric versus the best performing method
>
> 3. On Exchange, it is true that some of the simpler benchmarks (e.g. ETS) clearly outperform VQ-TR, but this also holds for all transformer-based methods in Table 1. This is because the Exchange dataset is a time series of currency exchange rates, which is more or less a random walk with extremely little predictive information. Therefore, it is difficult to outperform simple classical methods like ETS on this kind of dataset. Despite this, we do believe it is significant that, although transformer-based approaches do not seem to be ideal for datasets like Exchange, VQ-TR still outperforms the other transformer-based methods here (which we believe also validates our claim about the regularizing benefit of vector quantization!)
>
> It was not our intention to hide less favorable results in the appendix, rather for space reasons we decided to focus in the main paper on the most relevant comparisons against other transformer-based approaches. That being said, with the exception of the Exchange dataset, as explained above, the same kinds of trends that hold in Table 1 also hold in Table 3, and we believe that what we presented in the main paper is accurate and not misleading. However, we understand that the performance on Exchange requires some extra explanation and context, which we will add.
>
> # Choice of Output Distributions
>
> In order to have a more direct one-to-one comparison versus VQ-AR, for each dataset we used the same kind of distribution head for VQ-TR as the best performing distribution head of VQ-AR. In general, we used -t for datasets with continuous data, and -nb for datasets with count based data, with the exception that we used -iqn for Solar and Wikipedia (because this is what performed best for VQ-AR). In addition, we used different distribution heads for different datasets in order to highlight and reinforce the fact that the output head can be changed within the VQ-TR framework. Of course, one could perform explicit hyperparameter optimization on the choice of distribution head (which we did not do for the above reasons), and if we did this the performance of VQ-TR would potentially be even better. We will clarify this.
>
> In addition, for the Taxi dataset VQ-TR used a negative binomial head (same as VQ-AR), the -t suffix for Taxi is a typo that we will fix for the final paper.
>
> # Why does TFT have a Shorter Running Time
>
> The reason for this is that, unlike the other transformer-based methods, TFT by design only has a single self-attention module. Therefore, hyperparameter changes to increase the complexity/capacity of the model can only go towards layers outside of the attention mechanism, so no matter what hyperparameters are chosen it is always relatively fast compared with the other transformer-based baselines. That being said, it still has quadratic in context length scaling. Note that our comparison in Figure 2 does not show the asymptotic scaling of computational cost as context length increases, rather we show what the computation cost looks like in practice for a reasonably typical context length on this dataset. Also, it should be noted that although TFT is often fast in practice, we find that it performs relatively poorly on many datasets.

---

> > ### Comment · Reviewer_XX8d · 2023-11-16
> >
> > Thank you very much for your detailed comments. I concur with most points. I stand by my original statement of "motivation, exposition and the key idea follow closely from VQ-AR" but that the computational benefits are much more significant in the proposed architecture. I would also still urge the authors to discuss the nature of experimental results clearly in the main paper, that the results can be mixed based on the nature of the dataset.
> >
> > The authors have sufficiently addressed my concerns. I believe the potential benefits of the proposed method are far reaching and the paper is of interest to the ML community also outside time series forecasting. I will therefore revise my review.

---

### Author Response · Authors · 2023-11-16
**Response RE VQ-TR vs VQ-AR**

The relationship between VQ-TR and VQ-AR is *not* as simple as suggested by reviewers XX8d and JnRg. Indeed, our novel VQ-TR model is not given by simply replacing the RNN mechanism with transformers in the VQ-AR model. For reference, VQ-AR works as follows (see Figure 1 in Rasul et al. (2022) for a visual description):

1. Run an encoder RNN over the raw time series data
2. Pass the outputs of the encoder RNN at each time step through a VQ module
3. Run the decoder RNN over the quantized encoder outputs

This means that, if we were to just replace the RNNs with transformers in the VQ-AR model, the encoder transformer would be run over non-quantized vectors, and only the decoder transformer would be run over quantized vectors. Therefore, the quadratic complexity in the sequence length problem would still be present in the encoder part of the model, and so we would still have similar issues with scaling to long context lengths and general computational inefficiency.

Instead, what we propose to do with our VQ-TR model is place the VQ module *within* the transformer architecture for both the encoder transformers, which is completely novel to the best of our knowledge. We firmly believe that it is not at all obvious how to go about doing this effectively and that this is a very non-trivial extension of existing time series forecasting methods.

However, all of that being said, we acknowledge that the novelty of VQ-TR relative to VQ-AR was not as clear as it could have been and that the paper unintentionally gave the impression that it is a minor extension of this existing method. We will fix this and make the presentation of VQ-TR versus VQ-AR more clear, along the lines of the above.

EDIT:
Adding a little more detail here, there are potentially many ways of incorporating vector-quantization, or something similar, into the transformer architecture in order to achieve our computational efficiency goals. The particular novel approach we took in doing this was not random/arbitrary, but rather justified by a novel theoretical result. Specifically, we provided a bound on the error caused by approximation of the query vectors in the self-attention mechanism, and explained that this bound corresponds to the k-means loss function that is implicitly minimized by vector quantization (see our Theorem 3.1 and subsequent discussion.) This directly motivated exactly where/how we placed VQ modules within the transformer architecture (i.e. that VQ is applied to the query vectors in each self-attention layer). We believe that this theoretical justification makes our approach for solving this problem particularly interesting/novel.

---

### Author Response · Authors · 2023-11-16
**Response RE Codebook Sizes**

It is important to consider that the codebook size experiments that we performed are based on averaging over only 5 runs. Furthermore, the test performance itself is a noisy measure of the true out of sample performance due to the finite sample of data. Therefore, there is some inherent uncertainty in the mean predictions that we plotted, and these should *not* be interpreted to reflect an underlying chaotic or unstable dependence on codebook size. In addition, we note that while the variances in Figure 3 may look large at first glance, this is only because of the very narrow y-axis ranges used in these plots, as well as the fact that the mean effects of changing codebook size are also small.

When considering the degree to which VQ-TR is sensitive to codebook size optimization, it is important to note that the scale of variation seen in these plots is very small compared with the differences in CRPS values observed for different methods in Table 1. For example, with the Taxi dataset we see CRPS ranging from approx. 0.29 to 0.305 in Figure 3; even the most pessimistic end of this range is barely any worse than the most competitive benchmark methods (CRPS=0.301), and this is despite the fact that the comparison favors the benchmarks, since we are comparing VQ-TR performance without any dataset-specific hyperparameter optimization (other than codebook size) to benchmark performance with hyperparameter optimization (as seen in Table 1, when we do full hyperparameter optimization for VQ-TR on Taxi we get CRPS=0.281). Similar results also hold for the other datasets, which any reader can easily verify. In other words, although we see some interesting trends in Figure 3, on the whole VQ-TR is not very sensitive to the codebook size, and performs competitively even when codebook sizes are not well optimized. We will make this more clear in the final paper.

Regarding the trends or lack thereof that are evident in Figure 3, we agree that there are not very clear trends in all cases, and indeed we do not claim that there is a single consistent strong trend for all datasets. Rather, as noted in our paper, we see some interesting trends in some cases, predominantly for the Taxi and Traffic datasets. For both of these, two things are apparent: (1) the best performance is clearly obtained with very small codebook sizes; and (2) the variance of CRPS is much higher with very large codebook sizes. Together, these two observations strongly suggest an overfitting phenomenon for large codebooks, and that in these cases using small codebooks has a beneficial regularizing effect. Furthermore, this hypothesis is backed up by the nature of these two datasets: for both, we expect that the high-dimensional collection of time series should have a very simple, low-dimensional latent structure, driven by the general shared pattern of traffic / transport demand for different times of day, which increases the risk of overfitting. In other words, for the exact datasets where, given domain knowledge, we would expect regularization to be most important, we indeed see that VQ-TR benefits significantly from smaller codebook sizes. This is strong evidence that the VQ module can have a beneficial regularizing effect. We will make this discussion more clear in the final paper.

Finally, regarding the cost of optimizing the codebook size, we first note that as explained above, VQ-TR is not as sensitive to codebook size as Figure 3 might suggest at first glance, so in very computationally limited settings one could justify doing little or no optimization of codebook size. That being said, we find that even with relatively large codebook sizes the computational cost of VQ-TR does not increase overly much. For example, under the same conditions as in Figure 2, we see the following computational cost for VQ-TR with J=25 (what was reported in Figure 2) versus J=245 (the largest codebook size we experimented with):

- J=25: runtime = 71s, memory = 0.77Gb
- J=245: runtime = 98s, memory = 1.78GB

This is compared e.g. with runtime and memory usage of 91s / 2.40Gb for vanilla transformer. Indeed, although we increased the codebook size by a factor of 10, the running time and memory usage only increased by a relatively modest amount, with computational cost on par with most other methods. Furthermore, as evidenced in Figure 3, VQ-TR does not need codebook sizes this large for (approximately) optimal performance, so in practice, we could restrict hyperparameter optimization to lower values (J<100) where computational cost is much lower. We will include the additional result of VQ-TR with large codebook size in the final paper, and add a discussion on codebook size optimization along these lines.

---

### Author Response · Authors · 2023-11-16
**Response RE Context Window Sizes**

We consider the context length to be a hyperparameter, and it is separately optimized for each method/dataset in order to maximize performance. In general, longer context windows might cause overfitting, and shorter context windows might cause underfitting, and the optimal balance of these considerations may be different for each method, and is decided in a data-driven way via hyperparameter optimization. We note that, despite the fact that VQ-TR can better handle longer windows computationally, it was *not* given a head start with larger context windows. Rather, all methods were given the same choices of context window lengths for hyperparameter optimization.

---

### Meta-Review · Area_Chair_gRrC · 2023-12-10

**Metareview:**

In this paper, the authors propose a novel architecture for training transformers for time series. The main contributions are using vector quantization of the inputs and the capacity to do probabilistic estimates. The paper reads well, and during the discussion, the authors and the reviewers engaged in a revealing back-and-forth that improved the paper.

One of the limitations of the paper is the robustness results for the VQ before the transformer architecture is applied. There is almost no difference in performance and no degradation when the codebook only has one vector. For a very low number of vectors, I should expect not to have sufficient diversity in the input to the transformer and a degraded solution. This would need to be addressed and explained in the final version of the paper.

**Justification For Why Not Higher Score:**

The paper is a solid contribution, but not a significant change in the state of the art.

**Justification For Why Not Lower Score:**

The paper could be rejected. The method's performance with the number of quantized vectors does not seem to matter, which is odd, because the authors change the value from 1 to a few hundred.

---

### Decision · Program_Chairs · 2024-01-16

Accept (poster)